# Elucidation and Regulation of Tyrosine Kinase Inhibitor Resistance in Renal Cell Carcinoma Cells from the Perspective of Glutamine Metabolism

**DOI:** 10.3390/metabo14030170

**Published:** 2024-03-19

**Authors:** Kento Morozumi, Yoshihide Kawasaki, Tomonori Sato, Masamitsu Maekawa, Shinya Takasaki, Shuichi Shimada, Takanari Sakai, Shinichi Yamashita, Nariyasu Mano, Akihiro Ito

**Affiliations:** 1Department of Urology, Tohoku University Graduate School of Medicine, Sendai, Miyagi 980-8574, Japan; ken-morozm@uro.med.tohoku.ac.jp (K.M.); tomonori4659@uro.med.tohoku.ac.jp (T.S.); shimapp@uro.med.tohoku.ac.jp (S.S.); taka_sa0423@uro.med.tohoku.ac.jp (T.S.); yamashita@uro.med.tohoku.ac.jp (S.Y.); itoaki@uro.med.tohoku.ac.jp (A.I.); 2Department of Pharmaceutical Sciences, Tohoku University Hospital, Sendai, Miyagi 980-8574, Japan; masamitsu.maekawa.a2@tohoku.ac.jp (M.M.); takasaki_shinya@hosp.tohoku.ac.jp (S.T.); mano@hosp.tohoku.ac.jp (N.M.)

**Keywords:** glutamine metabolism, PTEN, renal cell carcinoma, TKI resistance, VEGFR, VEGF signaling

## Abstract

Tyrosine kinase inhibitors (TKIs) play a crucial role in the treatment of advanced renal cell carcinoma (RCC). However, there is a lack of useful biomarkers for assessing treatment efficacy. Through urinary metabolite analysis, we identified the metabolites and pathways involved in TKI resistance and elucidated the mechanism of TKI resistance. To verify the involvement of the identified metabolites obtained from urine metabolite analysis, we established sunitinib-resistant RCC cells and elucidated the antitumor effects of controlling the identified metabolic pathways in sunitinib-resistant RCC cells. Through the analysis of VEGFR signaling, we aimed to explore the mechanisms underlying the antitumor effects of metabolic control. Glutamine metabolism has emerged as a significant pathway in urinary metabolite analyses. In vitro and in vivo studies have revealed the antitumor effects of sunitinib-resistant RCC cells via knockdown of glutamine transporters. Furthermore, this antitumor effect is mediated by the control of VEGFR signaling via PTEN. Our findings highlight the involvement of glutamine metabolism in the prognosis and sunitinib resistance in patients with advanced RCC. Additionally, the regulating glutamine metabolism resulted in antitumor effects through sunitinib re-sensitivity in sunitinib-resistant RCC. Our results are expected to contribute to the more effective utilization of TKIs with further improvements in prognosis through current drug therapies.

## 1. Introduction

Although the majority of renal cell carcinoma (RCC) patients are diagnosed at early stages, 20% of RCC patients develop metastases and 30% of initial localized RCC progresses to metastatic disease after definitive therapy [1,2]. Recently, significant progress has been made in the treatment of metastatic RCC using tyrosine kinase inhibitors (TKIs) and immune checkpoint inhibitors (ICIs) [3]. Several clinical trials have demonstrated striking improvements in patient survival by combining multiple therapies based on TKIs and ICIs [4,5,6,7]. These agents have proven to be effective in improving patients’ outcomes; however, drug resistance and disease progression remain inevitable in many cases when combining multiple therapies. After combination therapy, the effective use of sequential TKI therapy is essential for improving prognosis.

Metabolomics is a comprehensive analysis that measures a wide range of metabolites and allows real-time quantification of changes in cancer metabolism [8]. Measuring changes in metabolites at various stages of cancer progression is a promising technique for discovering new therapeutic targets [9]. Among several previously identified glutamine transporters, ASCT2 has been reported to be specifically expressed in cancer cells [10]. Our previous in vitro study indicated that ASCT2 expression and glutamine metabolism were significantly increased in sunitinib-resistant RCC cells [10]. However, we did not conduct in vivo studies or assess clinical specimens. Recently, regulating glutamine metabolism has become key to overcoming TKI-resistant RCC in the era of combination therapies [11,12,13]. However, the anticancer effect against TKI-resistant RCC [14] and the mechanism of antitumor effects by regulating glutamine metabolism remain unclear [15].

Overcoming TKI resistance by regulating metabolism would improve the prognosis of patients with advanced RCC. The purpose of this study was to investigate the metabolism of TKI-resistant RCC and to clarify its antitumor effect and mechanism of regulating glutamine metabolism in vitro and in vivo.

## 2. Materials and Methods

### 2.1. Ethics Approval and Consent to Participate

This study was approved by the ethics review board of Tohoku University School of Medicine (authorization number: 2019-1-749). This study was conducted in accordance with the principles of the Declaration of Helsinki, and the animal experiments were approved by the Animal Care and Experimentation Committee of the Tohoku University Graduate School of Medicine (2018MdA-146).

### 2.2. Patients’ Background and Their Urine Sample Analysis for LC-MS/MS Analysis 

All LC-MS/MS analyses were performed using a LC-MS-8050 triple quadrupole tandem spectrometer coupled with a Nexera X2 UHPLC system (Shimadzu, Kyoto, Japan) and Lab Solutions software ver. 5.118 (Shimadzu, Kyoto, Japan). These conditions, along with the preparation of calibration standards and internal standards, were described in our previous report [10]. 

We compiled data from 8 patients treated at our institution with sunitinib as a first-line therapy for metastatic RCC between November 2017 and December 2018. Pre-sunitinib urine samples were collected on the day before the first administration and post-sunitinib samples before the 2nd-line therapy. At the time of targeted metabolomics analysis, 25 µL of each urine sample was combined with an equal volume of the given internal standard (IS), along with 200 µL of acetonitrile. The mixture was vortexed for 5 s and centrifuged at 15,000× *g* for 5 min at 4 °C. Aliquots (120 µL each) of the supernatant were transferred to separate 1.5 mL microcentrifuge tubes (cat. no. 509-GRD-Q; Thermo Fisher Scientific, MA, USA) and evaporated under reduced pressure for 1 h. Separate aliquots from a given sample were reconstituted with 20 µL of a 75:25 (*v*:*v*) mixture of acetonitrile and water. Subsequently, 20 μL of each solution was injected into the relevant analytical system. Clinical information and urine test results are presented as the mean and standard deviation, and were compared using Student’s t-test using the JMP Pro 15 software ver. 1.53e (SAS Institute, Cary, NC, USA).

Creatinine concentrations in all urine samples were measured using an enzymatic protocol with a CRE-CL commercial kit (Serotec, Sapporo, Japan). Absorbance was measured using an Infinite 200 Pro microplate reader (Tecan, Mennedorf, Switzerland). The measured metabolite urinary concentrations (µM) were adjusted according to the urinary creatinine concentration (mM) of the respective samples.

### 2.3. Cell Lines, Culture Conditions, and WST Assay

Human renal cell carcinoma cell lines (786-O (CVCL_1051), ACHN (CVCL_1067), and Caki-1 (CVCL_0234)) were purchased from the American Type Culture Collection (ATCC; Manassas, VA, USA) and maintained in culture media. Experiments on the majority of cell lines were conducted within 3–6 months, and 10 passages of purchase from the ATCC. Each cell line was seeded at 2 × 10^6^ cells/dish for subsequent analysis. Sunitinib-resistant RCC cells were generated by growing parental sunitinib-sensitive RCC cell lines (786-O, ACHN, and Caki-1) successively treated with increasing concentrations of sunitinib (S1042, Selleck, Houston, TX, USA) up to 10 µM. After continuous culture in complete medium supplemented with 10 µM sunitinib for >20 passages, these cells were used as sunitinib-resistant RCC cell lines (786-SR, ACHN-SR, and Caki-SR) and maintained in a medium containing 10 µM sunitinib. All experiments were performed using mycoplasma-free cells and repeated three times. All cell lines were certified using the ATCC STR database.

A WST assay was performed to measure the cytotoxicity of sunitinib. Sunitinib-sensitive RCC cells and sunitinib-resistant RCC cells were seeded onto a 96-well plate at a density of 5.0 × 10^3^ cells/well. After 24 h incubation, the culture medium was removed and the cells were treated with fresh medium containing different concentrations of sunitinib dissolved in dimethyl sulfoxide for 48 h. The number of cells was analyzed by the WST assay using the Cell Counting Kit-8 (Dojindo, Kumamoto, Japan). Absorbance was measured at 450 nm using a microplate spectrophotometer (Multiskan GO, Thermo Fisher Scientific, Waltham, MA, USA). The 50% cell growth inhibitory concentration (IC^50^) of each compound was determined.

### 2.4. Regulating Glutamine Metabolism

We evaluated glutamine metabolism pathways following ASCT2 blockade. ASCT2 blockade was performed via ASCT2 knockdown (KD) using siRNA transfection. ASCT2 siRNAs were purchased from Thermo Fisher Scientific (USA). The sequences of the siRNAs were as follows:siRNA#1: UUGAAGAAGCGGAUAAGCAGCUCCCsiRNA#2: UUUACGAAGUCCAAGGACUGCUGGC

SiRNA transfection was performed using Lipofectamine RNAiMAX (Thermo Fisher Scientific, USA). ASCT2 KD cells were named 786-OK, 786-SRK, Caki-K, Caki-SRK, ACHN-K, and ACHN-SRK.

### 2.5. Cell Proliferation, Wound-Healing, and Two-Chamber Assay

To compare the in vitro proliferation differences between sunitinib-sensitive cells, resistant cells, and ASCT2 KD cells, 1 × 10^4^ cells of each cell line were seeded with 5 µM sunitinib into each well of 24-well plates. The number of cells in each cell line was assessed daily in triplicate, using the Cell Counting Kit-8 by an Infinite 200 fluorescence plate reader (Tecan, Maninder, Switzerland).

For the wound-healing assay, cells were seeded into each well of 6-well plates in normal cell-growth medium and grown until confluence. Then, a 1 mL pipette tip was used to create a straight scratch line, simulating a wound. The medium was replaced with a medium containing 5 µM sunitinib. The area occupied by cells that migrated into the scratch area was measured using ImageJ software ver. 1.53e (National Institutes of Health, Bethesda, MD, USA). The ratio of the migration area to the scratch area was plotted. 

To conduct two-chamber assay, cell migration was assessed using a two-chamber assay with a Transwell 3422 (Corning, Corning, NY, USA). Approximately 5 × 10^4^ cells were plated in each cell culture insert in a serum-free medium containing 5 µM sunitinib. The bottom well contained a medium supplemented with 10% FBS and fibronectin (Corning, Corning, NY, USA). After 48 h, the bottom of the insert was stained with 1% crystal violet/D-PBS for 30 min, and the cells that invaded through the membrane to the lower surface were counted. 

### 2.6. In Vivo Study of Nude Mouse Model Grafted Sunitinib-Resistant Cells

Prior to conducting the in vivo study, we evaluated the period of time that ASCT2 KD remained effective when siRNA persisted in vitro. In quantitative real-time PCR (qRT-PCR), the expression of ASCT2 was reduced by 60–82% in 786-O and 56–97% in 786-SR at 3 weeks after SiRNA transfection. Therefore, an in vivo study was conducted within 3 weeks.

Female BALB/c-nu/nu mice (4–6-week-old) were purchased from CLEA Japan (Tokyo, Japan). 786-O, 786-OK, 786-SR, and 786-SRK cells were trypsinized and 2 × 106 cells were subcutaneously injected with 100 μL Matrigel (Corning, Corning, NY, USA). The tumor volume was calculated using the modified ellipsoid formula 1/2 (length × width^2^) after transplantation. In accordance with previous reports, 25 mg/kg/day was evaluated as a valid sunitinib dose [12]. The mice were divided into the following four experimental groups under sunitinib administration. Group A was subcutaneously transplanted with 786-O, Group B with 786-OK, Group C with 786-SR, and Group D with 786-SRK cells. Each group was treated for 2 weeks. Thereafter, the animals were euthanized, and the subcutaneous tumors were removed.

### 2.7. Chemicals and Reagents to Evaluate Intracellular Metabolites for LC-MS/MS Analysis

Mixtures of 100 µL IS and 100 µL of 50% water/acetonitrile were added to cell pellets in an in vitro study and to 20 mg of resected tumor in an in vivo study. These samples were prepared using the same protocol as that used for urine metabolites, as described above. The intracellular metabolite concentration in each cell was measured per 10^4^ cells in vitro and the concentration per 1 mg in the in vivo study was calculated. Statistical analyses were conducted using the Wilcoxon rank-sum test and differences between the groups were considered statistically significant at *p*  <  0.05.

### 2.8. RNA Extraction and Quantitative Real-Time Polymerase Chain Reaction

RNA was extracted from cell lysates using the acid guanidinium–phenol–chloroform method, and the total RNA was reverse transcribed into cDNA using the iScript™ cDNA Synthesis Kit (Bio-Rad Laboratories, Hercules, CA, USA) following the manufacturer’s protocol [12]. qRT-PCR was performed using the Dice Real Time System Thermal Cycler (TP900, Takara Bio Inc., Shiga, Japan) and SYBR Premix Ex Taq™ II (Takara, Shiga, Japan). The mRNA expression was evaluated by comparison with GAPDH expression.

### 2.9. Western Blot Analysis

Cells were lysed in radioimmunoprecipitation lysis buffer (Santa Cruz Biotechnology, Dallas, TX, USA) supplemented with phosphatase inhibitor cocktail (Kaygen, Irvine, CA, USA). Protein amounts in lysates were quantified using a Pierce BCA Protein Assay Kit (Thermo Fisher Scientific, Waltham, MA, USA). The separated proteins were then transferred to polyvinylidene difluoride (PVDF) membranes (Hybond-P PVDF, Amersham Biosciences, Little Chalfont, UK). After blocking and washing, PVDF membranes were incubated at 4 °C with shaking condition for 12 h to react with the primary antibody, and then incubated with secondary antibody at room temperature under shaking conditions for 60 min. Immunoreactive bands were developed using Clarity™ Western ECL Substrate (Bio-Rad Laboratories, Hercules, CA, USA), then detected and quantified using a ChemiDoc™ MP Imaging System (Bio-Rad Laboratories, Hercules, CA, USA). Protein expression was evaluated by comparison with GAPDH expression.

## 3. Results

### 3.1. Association between Urinary Glutamate Concentration and Prognosis in Patients Who Failed Sunitinib

Urine metabolites were evaluated in eight patients treated with sunitinib as a first-line treatment to compare the differences in metabolites before and after sunitinib resistance. We conducted accurate quantification of glutamine and glutamate based on previous research results. The mean pre-sunitinib urinary glutamine concentration was 1590.71 μmol/L/u-Cre and post-sunitinib 1653.63 μmol/L/u-Cre. The mean ratio of urinary glutamine concentration of post-sunitinib to that of pre-sunitinib was 9.99 and there was no significant difference (*p* = 0.96) (Figure 1a). The mean pre-sunitinib urinary glutamate concentration was 11.90 μmol/L/u-Cre and post-sunitinib 88.83 μmol/L/u-Cre. The mean ratio of urinary glutamate concentration of post-sunitinib to that of pre-sunitinib was 8.47 with a significant difference (*p* = 0.0013) (Figure 1a). Moreover, two patients with exceptionally high glutamate levels had poor prognosis with survival times of 4 and 7 months after sunitinib resistance, compared to the other six patients with survival times of 13–50 months (Figure 1b).

### 3.2. Expression of Glutamine Transporter in Three Established Sunitinib-Resistant Cell Lines

As shown by the WST, we successfully established resistant strains with 4.6–7.7-fold drug resistance (Table 1). Therefore, it was demonstrated that maintaining the concentration of sunitinib in the culture medium above 5 μM is reasonable for analyzing the viability of sunitinib-resistant cell lines. In all three cell lines, qRT-PCR and Western blotting also showed overexpression of ASCT2 in sunitinib-resistant cells compared to sunitinib-sensitive cells (Figure 2a). When comparing the expression of ASCT2 among sunitinib-sensitive cells, ASCT2 was found to be highly expressed in 786-O compared to that in Caki-1 and ACHN (Figure 2a). Sunitinib-resistant cells had higher intracellular concentrations of glutamine metabolism (glutamine, glutamate, and αKG) (Figure 2b). When comparing intracellular metabolite concentrations among sunitinib-sensitive cells, all three metabolites were higher in 786-O than in Caki-1 and ACHN. Glutamine concentration of 786-O, Caki-1, and ACHN cells were 629.89 μM, 127.64 μM, and 164.83 μM, whereas those of 786-SR, Caki-SR, and ACHN-SR were 2058.25 μM, 1273.46 μM, and 440.24 μM, respectively (Figure 2b). Glutamate concentration of 786-O, Caki-1, and ACHN cells were 72.49 μM, 52.78 μM, and 38.73 μM, while those of 786-SR, Caki-SR, and ACHN-SR were 148.25 μM, 116.50 μM, and 100.92 μM, respectively (Figure 2b). αKG concentration of 786-O, Caki-1, and ACHN cells were 24.44 μM, 7.81 μM, and 1.49 μM, whereas those of 786-SR, Caki-SR, and ACHN-SR were 34.43 μM, 25.87 μM, and 5.09 μM, respectively (Figure 2b).

### 3.3. Re-Sensitivity to Sunitinib Resulting in Antitumor Effects through Attenuation of Enhanced ASCT2 in Resistant Cells

We used two siRNAs and conducted experiments using siRNA #2 with a more efficient KD. In all cell lines, qRT-PCR and WB showed that ASCT2 expression was higher in sunitinib-resistant cells than in sunitinib-sensitive cells, and significantly decreased after ASCT2 KD was administered (Figure 3a).

First, 786-O, 786-OK, 786-SR, and 786-SRK were cultured with 5 µM sunitinib to evaluate their proliferation, migration, and invasion capacities. The proliferation assay showed that the proliferation capacity of 786-OK decreased by 41% compared to that of 786-O, and decreased by 45% in 786-SRK compared to that of 786-SR (Figure 3b). The wound-healing assay indicated that the migration capacity of 786-OK decreased by 65% than that of 786-O, and decreased by 78% in 786-SRK compared to that of 786-SR (Figure 3c). The two-chamber assay also showed that the invasion capacity of 786-OK decreased by 44% compared to that of 786-O cells, and decreased by 78% in 786-SRK compared to that of 786-SR (Figure 3d). When comparing the assay, migration, and invasion capacity, each had significantly lower values. Additionally, all cancer activities decreased significantly in 786-SRK compared to those in 786-OK.

In sunitinib-sensitive cells (Caki-1 and ACHN), in contrast to 786-O, almost all assays showed no extension of antitumor effect in ASCT2 KD (Figure 3b–d). In contrast, proliferation, migration, and invasion capacities were significantly decreased in sunitinib-resistant cells (Caki-SR and ACHN-SR). The proliferation assay showed that the proliferation capacity decreased by 67% in Caki-SRK compared to that in Caki-SR, and that of ACHN-SRK decreased by 74% compared to that in ACHN-SR (Figure 3b). The wound-healing assay indicated that the migration capacity decreased by 66% in Caki-SRK compared to that in Caki-SR, and that of ACHN-SRK decreased by 60% compared to that in ACHN-SR (Figure 3c). The two-chamber assay also showed that the invasion capacity decreased by 42% in Caki-SRK compared to that in Caki-SR, and that of ACHN-SRK decreased by 79% compared to that in ACHN-SR (Figure 3d).

### 3.4. Relationship between Glutamine Metabolism and VEGF in Sunitinib-Resistant Cells

When ASCT2 KD was administered, intracellular glutamine, glutamate, and αKG levels were decreased in all cells (Figure 4a, Table 2). Higher expression of both VEGFR1 and VEGFR2 was observed in sunitinib-resistant cells than in sunitinib-sensitive cells (Figure 4b). After ASCT2 KD was administered, VEGFR2 was significantly downregulated in both sunitinib-sensitive and sunitinib-resistant cells, whereas VEGFR1 was not (Figure 4b). Additionally, evaluation of vascular endothelial growth factor (VEGF) signaling revealed upregulation of phosphatase and tensin homolog (PTEN) expression. After ASCT2 KD was administered, phosphoinositide 3-kinase (PI3K) was suppressed at a rate of 10–95%, AKT was suppressed at a rate of 23–80%, and hypoxia inducible factor-1α (HIF1α) was suppressed at a rate of 55–85% (Figure 4b).

### 3.5. Re-Sensitivity to Sunitinib In Vivo by ASCT2 KD

In the in vitro study, stronger antitumor effects of ASCT2 KD were observed in 786-O and 786-SR than in other cell lines. Therefore, we conducted an in vivo study using 786-O and 786-SR to evaluate the effectiveness of ASCT2 KD beneath the vascular endothelium.

A reduction of 67% in tumor size between 786-O and 786-OK (Figure 5a) and a reduction of 97% in tumor size between 786-SR and 786-SRK was observed with sunitinib administration at the time the mice were euthanized (Figure 5a).

In the intracellular glutamine metabolism analysis, levels of glutamine and glutamate were significantly higher in 786-SR than in 786-O, and decreased by ASCT2 KD in both 786-O and 786-SR (Figure 5b). The mean glutamine concentration per 1 mg in Group A was 228.25 μM, 0.24 μM in Group B, 4255.22 μM in Group C, and 380.78 μM in Group D (Figure 5b). The mean glutamate concentration per 1 mg in Group A was 240.35 μM, 2.46 μM in Group B, 2740.04 μM in Group C, and 163.37 μM in Group D (Figure 5b). The mean αKG concentration per 1 mg in Group A was 0.00094 μM, 0.0012 μM in Group B, 0.021 μM in Group C, and 0.00051 μM in Group D (Figure 5b).

In an in vivo study with sunitinib, ASCT2 KD significantly decreased ASCT2 expression in qRT-PCR in both 786-O and 786-SR (Figure 5b). VEGFR1 was upregulated, and VEGFR2 was downregulated in ASCT2 KD cells (Figure 5b). Additionally, PI3K/AKT was suppressed, and PTEN was upregulated in ASCT2 KD cells (Figure 5c).

## 4. Discussion

Although TKIs are an effective treatment for RCC, there has recently been a revolutionary development of ICIs [14]. Despite the development of RCC therapy such as the combination of TKI plus ICIs or dual ICIs, TKIs are still one of the most prominent drugs. Since the majority of patients ultimately experience disease progression caused by drug resistance, sequential therapy with TKIs is needed for most patients with advanced RCC. However, we do not have sufficient data to guide optimal management after treatment failure with these combination therapies [14].

Upregulation of glutamine metabolism has been reported to play a crucial role in the progression toward higher malignancy in several cancers [10,11,12,13,15]. We reported that glutamine metabolism had also been reported to have a pivotal role for RCC [10]. In this study, we showed precise quantitative measurements of urine metabolites in eight patients treated with sunitinib as first-line therapy and identified glutamate as an essential metabolite associated with sunitinib resistance. Additionally, poor prognosis, and resistance to second-line treatment are associated with urinary glutamate levels. In our previous research, the accurate quantification of urinary glutamate reflected the malignant and recurrent status of RCC after definitive therapy [16,17]. Elevated glutamate levels have also been reported to be associated with poor prognosis [18]. Therefore, urinary glutamate has the potential to become a useful biomarker for sunitinib resistance, and regulating glutamine metabolism could be a therapeutic target. Wang et al. reported that blocking ASCT2 to prevent glutamine metabolism uptake has been shown to successfully prevent tumor cell proliferation in several cancers [19,20]. ASCT2 expression level has also been reported to be related to the therapeutic effect of regulating glutamine metabolism [19,21]. Our previous study reported a significant enhancement in ASCT2 expression in TKI-resistant RCC cells [10].

In this study, we highlighted that ASCT2 blockade had antitumor effects in both sunitinib-sensitive and sunitinib-resistant cells, and we found that the efficacy was greater in sunitinib-resistant cells than in sunitinib-sensitive cells. In vitro, we observed a decrease in proliferation, migration, and invasion ability by regulating glutamine metabolism in all three sunitinib-resistant cell lines. The migration and invasion abilities decreased more than the proliferation ability. Metabolomics analysis showed that the levels of glutamine, glutamate, and αKG were significantly higher in sunitinib-resistant cells than in sunitinib-sensitive cells, and decreased after the administration of ASCT2 KD in both sunitinib-sensitive and sunitinib-resistant cells. We also confirmed the antitumor effect of ASCT2 KD in vivo using 786-O and 786-SR, which were the most therapeutically effective among the three cell lines. ASCT2 blockade also had antitumor effects, and its efficacy was remarkable in sunitinib-resistant cells. The expression of ASCT2 in qRT-PCR and changes in glutamine metabolism in metabolomics analysis were also higher in sunitinib-resistant cells in vitro.

We found that ASCT2 KD led to PTEN activation. PTEN acts as a negative regulator for the VEGF signaling pathway and MYC-driven tumor genesis [22,23]. PTEN loss has been reported to be associated with enhanced production of energy metabolites, such as glutamine metabolism and glycolysis [24,25,26]. Hamadneh et al. reported that higher MYC expression caused by PTEN loss required more glutamine synthesis in cancer cells [24]. Wang et al. also reported that PTEN-deficiency might promote glutamine metabolism [26]. These reports led us to believe that controlling PTEN by regulating mitochondrial metabolism would be effective in cancer treatment [22]. In summary, ASCT2 KD leads to PTEN activation in sunitinib-resistant cells, which could lead to re-sensitivity to sunitinib. Moreover, overexpression of VEGF signaling (PI3K/AKT/HIF1α) was observed in sunitinib-resistant cells, and this signaling was suppressed by ASCT2 KD. The intracellular glutamine pool is critical for sustaining the activation of VEGF signaling and is a master regulator of cell growth, protein translation, and apoptosis [10]. Glutamine metabolism has also been reported to be associated with the PI3K/AKT signaling pathway, which leads to the activation of HIF [19]. As can be seen from our study and these reports, ASCT2 KD could lead to the suppression of upregulated VEGF signaling directly in TKI resistance.

Although ASCT2 interacts with other proteins, such as LAT1 or EGFR, the relationship between ASCT2 and VEGFR is unknown [26]. In this study, a higher expression of VEGFR1 and VEGFR2 was observed in TKI-resistant cells than in sunitinib-sensitive cells, and VEGFR2 was significantly downregulated by ASCT2 KD in both sunitinib-sensitive cells and sunitinib-resistant cells. VEGFR2 is the chief pathway in vasculogenesis and angiogenesis in cancer, and inhibition of this pathway has been reported to be crucial for RCC patient survival [27,28]. Zhou et al. reported that sunitinib resistance resulted in the overexpression of VEGFR2 and upregulation of the VEGF signaling pathways [25]. Therefore, ASCT2 KD can bring about re-sensitivity to sunitinib, caused by the suppression of VEGF signaling and VEGFR2 expression. In brief, in TKI-sensitive status, the VEGF signaling pathway can be controlled by TKI (Figure 6a). However, TKI cannot fully suppress overexpressed VEGFR2 in TKI-resistant status, leading to continuous activation of VEGF signaling (Figure 6b). When regulating glutamine metabolism by ASCT2 KD, re-sensitivity to TKI can be induced even after TKI resistance, by the downregulation of VEGFR2 and activation of PTEN with suppression of VEGF signaling (Figure 6c).

In sunitinib-sensitive cell lines, on the other hand, ASCT2 KD was only effective in 786-O, which had higher ASCT2 expression than Caki-1 and ACHN. Sato et al. indicated that RCC patients with high ASCT2 expression had a significantly poorer prognosis [10]. Thus, ASCT2 KD could be an effective treatment for RCC patients after treatment failure with sunitinib and for patients with high ASCT2 expression who have poor prognosis even if they are sunitinib-sensitive.

Although several specific drugs have been used to dysregulate glutamine metabolism, they are mainly divided into the following three groups: ASCT2, GLS, and GDH inhibitors. ASCT2 blockade involves adverse events (AEs) caused by glutamine deficiency [21]. Glutamine deficiency causes mesenchymal tissue differentiation disorders, inflammation, and multiple organ failure [29,30,31]. From these reports and this study, dysregulating glutamate could be crucial not only for treating TKI-resistant RCC but also for reducing unnecessary AEs. Inhibition of glutaminase, which converts glutamine to glutamate, is less toxic and shows no increase in AEs rates in clinical trials. Therefore, this agent may be suitable for clinical practice [20,21].

This study has the following limitations. First, the LC-MS measurement system, which uses our institutional settings, has not been externally validated. Second, patients were retrospectively enrolled, and only eight patients participated in this study. Third, TKI resistance was evaluated in three cell lines treated with only sunitinib. Fourth, the relationship between glutamine metabolism and VEGF signaling in the vascular endothelium remains unclear. Finally, we only evaluated ASCT2 blockade, although several specific drugs have been used to dysregulate glutamine metabolism [29,30,31]. 

## 5. Conclusions

In conclusion, despite these limitations, we revealed that regulating glutamine metabolism leads to re-sensitivity to TKI by activating PTEN and suppressing VEGFR2 and VEGF signaling. Glutamine metabolism could be a key metabolic pathway to overcome TKI-resistant RCC and improve the prognosis of patients with advanced RCC through more effective use of TKIs.

## Figures and Tables

**Figure 1 metabolites-14-00170-f001:**
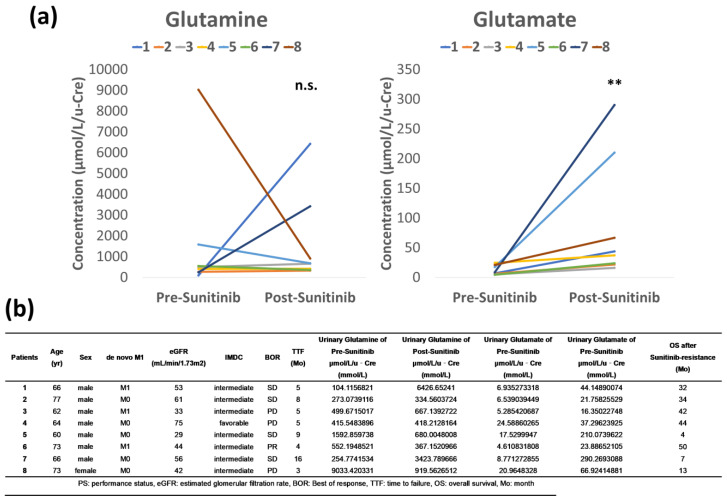
The change of glutamine metabolism in the urinary metabolites before and after sunitinib treatment and the background of patients who were treated with sunitinib as a first-line treatment. (**a**) Although the mean ratio of glutamine of the patients after experiencing sunitinib failure to those before they developed sunitinib failure was 9.99 with no significant difference (*p* = 0.96), the mean ratio of glutamate was 8.47 with a significant difference (*p* = 0.0013). (**b**) The background of patients who were treated with sunitinib as a first line treatment. ** *p* < 0.01, n.s.: not significant.

**Figure 2 metabolites-14-00170-f002:**
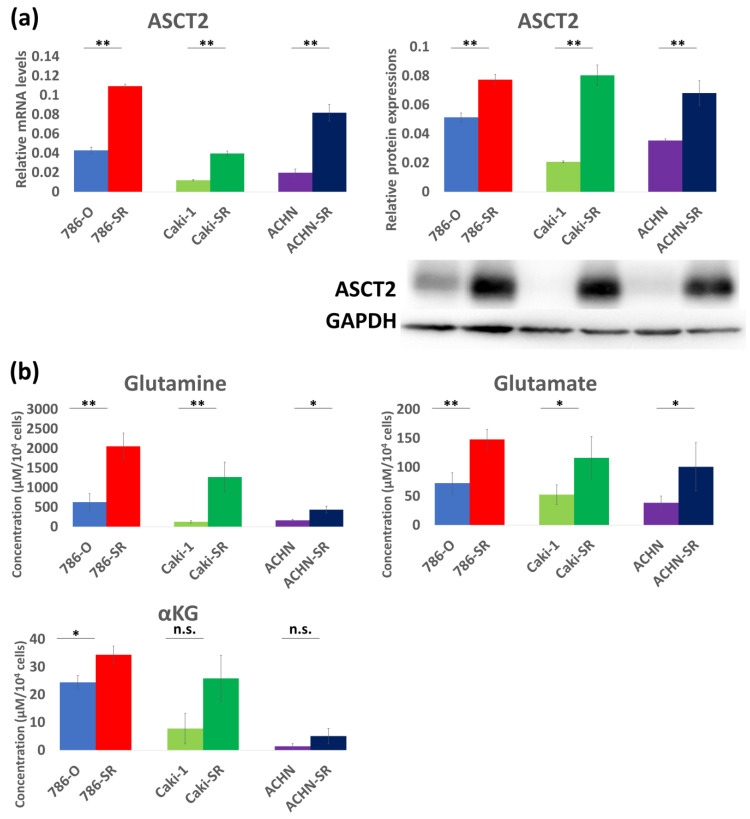
The change of glutamine metabolism in the in vitro study. (**a**) Sunitinib-resistant cells had significantly higher ASCT2 expression than sunitinib-sensitive cells. Among sunitinib-sensitive cells, ASCT2 was highly expressed in 786-O than the others in both RT-PCR and Western blotting. (**b**) Intracellular metabolite concentrations of 3 metabolites related to glutamine metabolism were significantly higher in sunitinib-resistant cells than in sunitinib-sensitive cells. When intracellular metabolite concentrations among sunitinib-sensitive cells were compared, all 3 metabolites were higher in 786-O than the others. * *p* < 0.05, ** *p* < 0.01, n.s.: not significant.

**Figure 3 metabolites-14-00170-f003:**
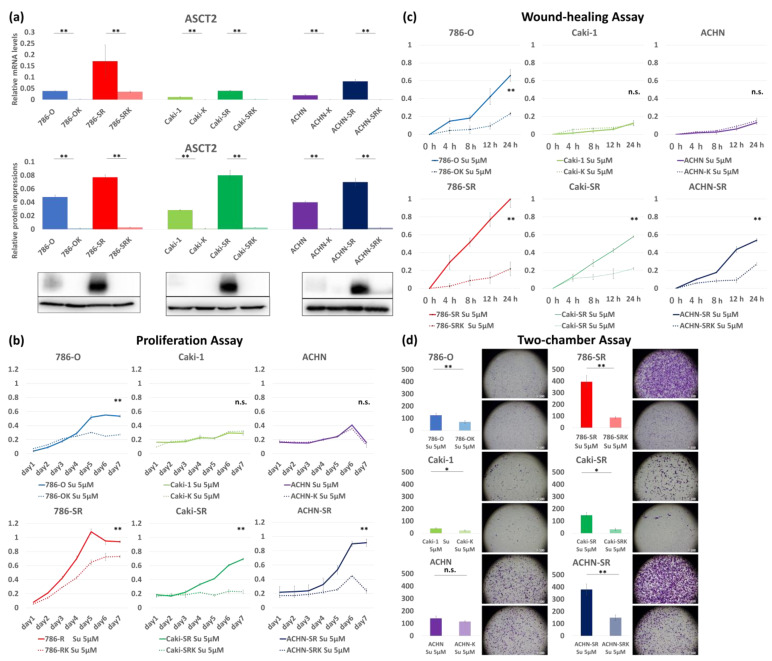
The evaluation of antitumor effects by ASCT2 KD in sunitinib-sensitive and sunitinib-resistant cells. (**a**) ASCT2 expression significantly decreased in both sunitinib-sensitive and sunitinib-resistant cells when ASCT2 KD was administered. (**b**) In the proliferation assay, proliferation capacity was significantly decreased by ASCT2 KD in all sunitinib-resistant cells. Among sunitinib-sensitive cells, 786-OK was the only cell line with decreased proliferation capacity. (**c**) In the wound-healing assay, invasion capacity was significantly decreased by ASCT2 KD in all sunitinib-resistant cells. Among sunitinib-sensitive cells, 786-OK was the only cell line with decreased invasion capacity. (**d**) In the two-chamber assay, migration capacity was significantly decreased by ASCT2 KD in all sunitinib-resistant cells. Among sunitinib-sensitive cells, 786-OK was the only cell line with decreased migration capacity. * *p* < 0.05, ** *p* < 0.01, n.s.: not significant.

**Figure 4 metabolites-14-00170-f004:**
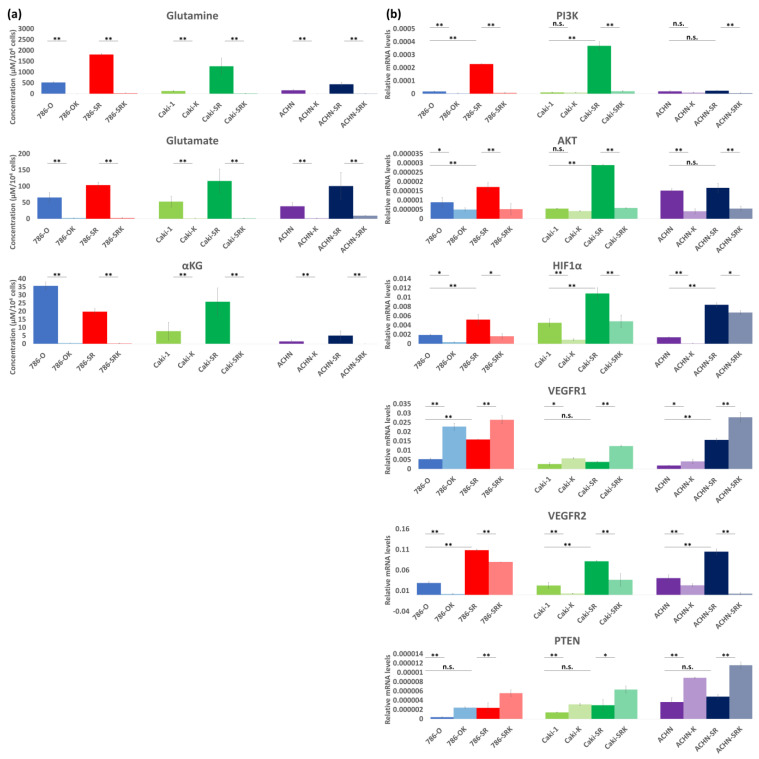
The evaluation of intracellular glutamine metabolism and expression of VEGFR and VEGF signaling (PI3K/AKT/HIF1α) after administration of ASCT2 KD. (**a**) Intracellular concentrations of glutamine, glutamate, and α-KG significantly decreased after ASCT2 KD was administered in both sunitinib-sensitive and sunitinib-resistant cells. (**b**) Higher expressions of both VEGFR and VEGF signaling were observed in sunitinib-resistant cells than in sunitinib-sensitive cells. After ASCT2 KD was administered, VEGFR2 and VEGF signaling were significantly downregulated. Additionally, ASCT2 KD led to upregulation of PTEN. * *p* < 0.05, ** *p* < 0.01, n.s.: not significant.

**Figure 5 metabolites-14-00170-f005:**
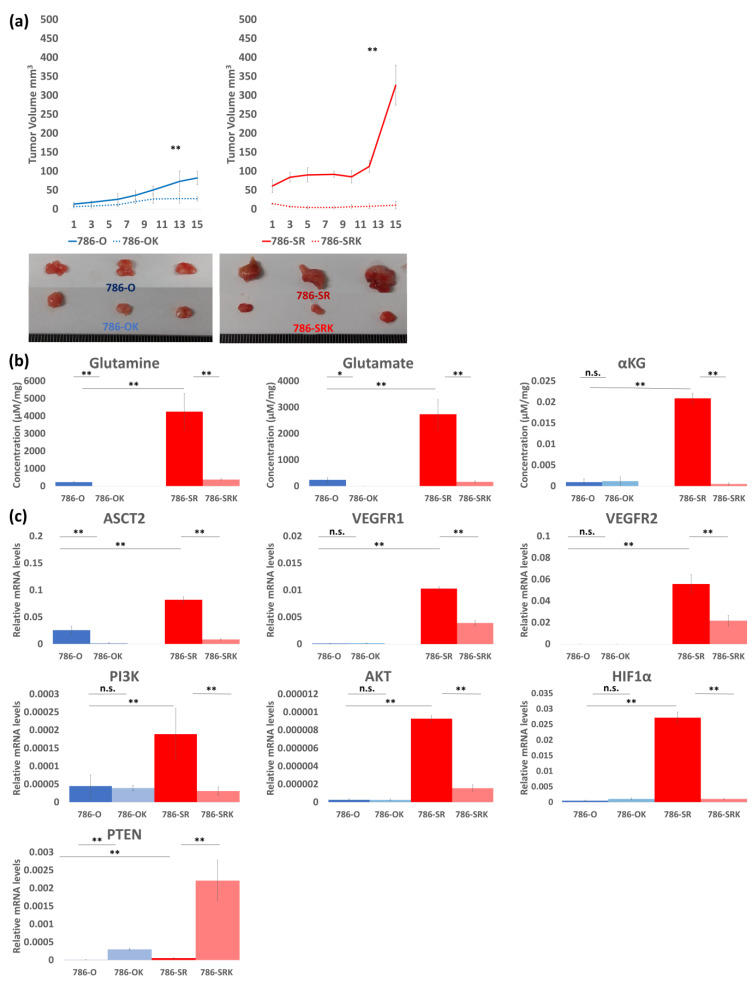
In vivo study conducted to evaluate antitumor effects by ASCT2 KD in 786-O and 786-SR. (**a**) Tumor growth curve indicated a reduction of 67% in tumor size between 786-O and 786-OK, and a reduction of 97% in tumor size between 786-SR and 786-SRK was observed with sunitinib administration at the time mice were euthanized. (**b**) The evaluation of intracellular glutamine metabolism in the in vivo study (**c**) The expression of ASCT2, VEGFR, PTEN, and VEGFR signaling (PI3K/AKT/HIF1α) in the in vivo study. * *p* < 0.05, ** *p* < 0.01, n.s.: not significant.

**Figure 6 metabolites-14-00170-f006:**
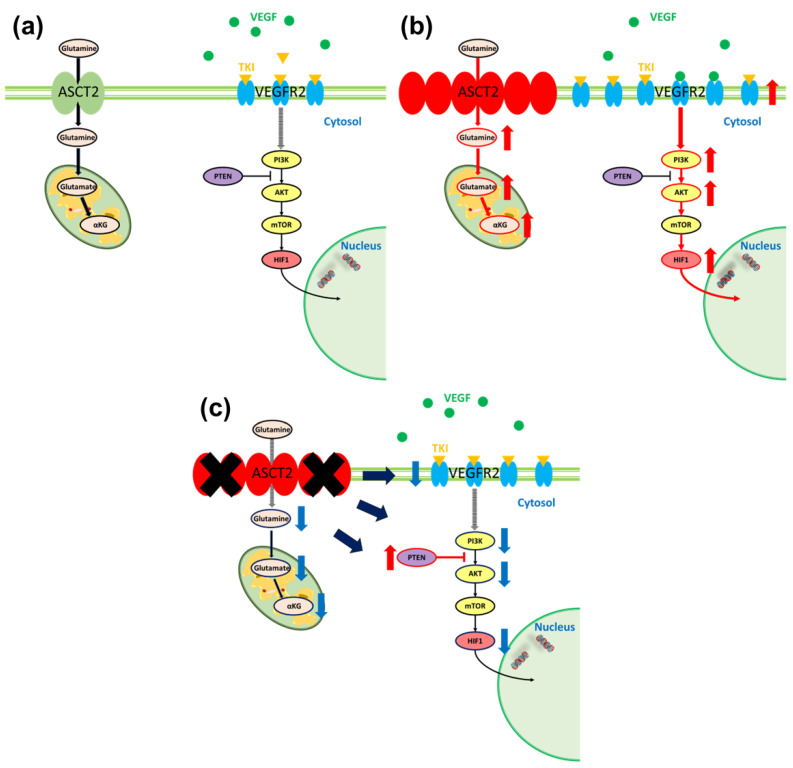
The proposed mechanism of re-sensitivity to TKI by regulation of glutamine metabolism. (**a**) During TKI-sensitivity, In TKI-sensitive cells, VEGF signaling pathway can be controlled by TKI. (**b**) TKI cannot fully suppress the overexpressed VEGFR2 during TKI-resistance in TKI-resistant counterparts, leading to a continuous activation of VEGF signaling. (**c**) When regulating glutamine metabolism by ASCT2 KD, re-sensitivity to TKI can be inducted achieved even after TKI-resistance, by the downregulation of VEGFR2, activation of PTEN and suppression of VEGF signaling.

**Table 1 metabolites-14-00170-t001:** IC^50^ to evaluate the growth inhibitory effect of sunitinib in sunitinib-sensitive cell lines and sunitinib-resistant cell lines.

IC^50^ (μM)	Sunitinib-Sensitive Cells	Sunitinib-Resistance Cells
	786-O: 4.8 (95%CI 2.1–10.8)	786-SR:22.1 (95%CI 15.0–29.8)
	Caki-1: 2.7 (95%CI 0.6–8.5)	Caki-SR: 14.8 (95%CI 8.7–19.6)
	ACHN: 2.1 (95%CI 0.9–6.0)	ACHN-SR: 2.1 (95%CI 9.5–20.1)

**Table 2 metabolites-14-00170-t002:** The comparison of intracellular metabolite concentrations among sunitinib-sensitive cells, sunitinib-resistant cells, and each ASCT2 KD cell line.

	Glutamine (μM)	Glutamate (μM)	αKG (μM)
**786-O** **786-OK**	629.89 5.18	72.49 2.54	24.44 0.43
**786-SR** **786-SRK**	2058.25 37.95	148.25 2.88	34.43 0.30
**Caki-1** **Caki-K**	127.64 3.48	52.78 1.35	7.81 0.0036
**Caki-SR** **Caki-SRK**	1273.46 24.54	116.50 1.61	25.87 0.0091
**ACHN** **ACHN-K**	164.83 3.98	38.73 1.85	1.49 0.0054
**ACHN-SR** **ACHN-SRK**	440.24 14.46	100.92 9.52	5.09 0.12

## Data Availability

The datasets generated and analyzed in the current study are available from the corresponding author on reasonable request.

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
