# Peer review of "Elucidation and Regulation of Tyrosine Kinase Inhibitor Resistance in Renal Cell Carcinoma Cells from the Perspective of Glutamine Metabolism"

_metabolites, 2024, doi:10.3390/metabo14030170_

Round 1

Reviewer 1 Report

Comments and Suggestions for Authors

The study by K. Morozumi et al. analyzed an important problem in cancer biology, namely, the resistance of renal carcinoma to the tyrosine kinase inhibitor sunitinib. The authors provided evidence in favor of a mechanistic link between the resistant phenotype and glutamine metabolism. In general, this finding is of fundamental importance and is attractive from the therapeutic perspective. 

To better evaluate this work I suggest the following revision:

1. Figures are hardly readable: the font is too small, so the text on the axes is virtually invisible. Please make figures less busy or anyway enlarge the captions.

2. Figure 6 (mentioned in Discussion) is missing in the current version. Please add. One may suggest that it is the scheme of mechanisms; once it is available the evaluation of the study can be finalized.

3. The authors discuss the role of PTEN (page ). 

Comments on the Quality of English Language

The text needs a thorough linguistic editing. 

Reviewer 2 Report

Comments and Suggestions for Authors

The manuscript titled "Elucidation and Regulation of Tyrosine Kinase Inhibitor Resistance in Renal Cancer Cells from the Perspective of Glutamine Metabolism" by Kento Morozumi et al. provides valuable insights into the mechanisms underlying TKI resistance in RCC. The authors successfully identified key metabolites and pathways associated with TKI resistance through urinary metabolite analysis. Their in vitro work with sunitinib-resistant RCC cells and subsequent investigation of the antitumor effects of controlling metabolic pathways further strengthens their findings. The study highlights the significance of targeting glutamine metabolism and its potential to enhance the efficacy of sunitinib (and potentially other drugs) in RCC chemotherapy. Considering the rigorous experimental design and compelling results, I recommend accepting this work in its current form.

Round 2

Reviewer 1 Report

Comments and Suggestions for Authors

The authors addressed my comments. Still, I dared to edit the legend to Figure 6. See my suggestions in red. 

Figure 6. The proposed mechanism of re-sensitivity to TKI by regulation of glutamine metabolism.

a) During TKI-sensitivity, In TKI-sensitive cells, VEGF signaling pathway can be controlled by TKI. b) TKI cannot fully suppress the overexpressed VEGFR2 during TKI-resistance in TKI-resistant counterparts, leading to a continuous activation of VEGF signaling. c) When regulating glutamine metabolism by ASCT2 KD, re-sensitivity to TKI can be inducted achieved  even after TKI-resistance, by the downregulation of VEGFR2, activation of PTEN and suppression of VEGF signaling.

Now the study can be recommended for publication. 

Comments on the Quality of English Language

The text should be thoroughly edited; this can be done by the professional expert during the production process.